# Genitourinary Tissue Engineering: Reconstruction and Research Models

**DOI:** 10.3390/bioengineering8070099

**Published:** 2021-07-13

**Authors:** Christophe Caneparo, David Brownell, Stéphane Chabaud, Stéphane Bolduc

**Affiliations:** 1Centre de Recherche en Organogénèse Expérimentale/LOEX, Regenerative Medicine Division, CHU de Québec-Université Laval Research Center, Québec, QC G1J 1Z4, Canada; christophe.caneparo@crchudequebec.ulaval.ca (C.C.); david.brownell@espci.fr (D.B.); Stephane.Chabaud@crchudequebec.ulaval.ca (S.C.); 2Division of Urology, Department of Surgery, CHU de Québec-Université Laval, Québec, QC G1V 4G2, Canada; 3Department of Surgery, Faculty of Medicine, Laval University, Québec, QC G1V 0A6, Canada

**Keywords:** tissue engineering, urethra, vagina, urology, gynaecology, epithelium

## Abstract

Tissue engineering is an emerging field of research that initially aimed to produce 3D tissues to bypass the lack of adequate tissues for the repair or replacement of deficient organs. The basis of tissue engineering protocols is to create scaffolds, which can have a synthetic or natural origin, seeded or not with cells. At the same time, more and more studies have indicated the low clinic translation rate of research realised using standard cell culture conditions, i.e., cells on plastic surfaces or using animal models that are too different from humans. New models are needed to mimic the 3D organisation of tissue and the cells themselves and the interaction between cells and the extracellular matrix. In this regard, urology and gynaecology fields are of particular interest. The urethra and vagina can be sites suffering from many pathologies without currently adequate treatment options. Due to the specific organisation of the human urethral/bladder and vaginal epithelium, current research models remain poorly representative. In this review, the anatomy, the current pathologies, and the treatments will be described before focusing on producing tissues and research models using tissue engineering. An emphasis is made on the self-assembly approach, which allows tissue production without the need for biomaterials.

## 1. Introduction

Despite the significant advances in medicine of the last decades, many illnesses are still challenging. Indeed, the treatments themselves can be insufficient, or their side effects may negatively impact the patient’s quality of life. From congenital anomalies to various injuries or illnesses that may develop, the list goes on and can range from a simple annoyance to a deadly threat. Affectations of the genitourinary tissues are no exception in this respect and can significantly contribute to undermining the social and emotional life of individuals [1,2,3,4,5]. In this context, a particular focus will be placed on the male urethra and the vagina. This review examines the anatomy, pathologies and treatments. Current treatments will be studied, as well as encountered problems and possible solutions offered by tissue engineering. The field of tissue engineering has emerged in the past decades, aiming to treat or replace damaged tissues. Various approaches will be described, and some of their applications either repair the urethra and the vagina or create study models. A new and exceptionally innovative approach will then be put forward: The self-assembly technique [6].

## 2. Male Genitourinary Tissue: Focus on the Urethra

The blood is filtered through the kidneys to produce urine which contains toxic waste. Urine is carried through the ureters down to the bladder, where it is stored until its excretion via the urethra. The latter is also used during reproduction to transport semen.

### 2.1. Anatomy

The 15 to 20 centimetres of the male urethra are divided into several sections from the bladder neck to the meatus, and described as the prostatic, membranous, bulbar, pendulous and fossa navicularis sections of the urethra. The two first parts constitute the posterior urethra, whereas the following are known as the anterior urethra. The urethra is lined by an epithelium whose organisation varies by location. The posterior urethra is characterised by a transitional epithelium followed by a pseudostratified epithelium. Both are very similar and are known as urothelium. Finally, the squamous epithelium is found in the fossa navicularis, roughly similar to the oral mucosa [7]. The urothelium consists of three elements. The basal layer contains most of the progenitor/stem cells allowing for normal turnover of the epithelium. On the basal layer lies several layers containing intermediate cells with a potential of division depending on their level of differentiation. The progenitor/stem cells present in these layers are those implicated in repair after injuries. Finally, the superficial layer contains flattened, sometimes binucleated, terminally differentiated cells. These cells are called umbrella cells and are mainly responsible for the barrier function against urine [8]. At least in part, the basal and the intermediate cells are directly connected to the basal lamina, which separates the urothelium from the lamina propria, the underlying connective tissue. Like other lower urinary tract structures, the urethra contains three layers of muscles: Internal and external muscle layers are longitudinal, whereas the medial muscle layer is circular. Internal and medial muscle fibres are extensions of the muscle fibres of the bladder detrusor, while the outer longitudinal fibres are composed of striated muscles.

### 2.2. Pathologies

Surgical reconstructions are often needed to restore the normal function of the genitourinary system of patients affected by congenital or acquired anomalies. Among them, hypospadias and urethral stricture are the most common pathologies seen by urologists. One in 250 newborn male is affected by hypospadias [9,10], a penile malformation representing 73% of all congenital penile anomalies [11]. Several studies have reported an increasing prevalence of hypospadias in men [12,13,14,15,16]. Its prevalence varies by region: It was lower in Asia and higher in North America [17]. Studies have estimated that the genetic component of hypospadias is between 57 and 77% risk of inheriting, coming through both maternal and paternal sides [18]. Specific mutated genes were brought to light that can influence the apparition of hypospadias, such as Wilms’ tumour 1 or androgen receptor genes [19]. Because the risks for siblings and sons from the same family are similar, genetic and shared environmental factors, such as endocrine disruptor exposition, are expected to play a significant role in familial hypospadias [20]. Androgens and oestrogens play a critical role in genital development. Animal models have already shown the impact of exposure to synthetic oestrogens on hypospadias prevalence.

Additionally, maternal exposure to these components and certain drugs, such as diethylstilbestrol or valproate, has been associated with a higher risk of hypospadias [21]. Hypertension, oligohydramnios and preterm delivery are linked to severe hypospadias, suggesting a significant role of underlying placental insufficiency potentially through inadequate levels of human chorionic gonadotropin in the foetus [22]. Hypospadias is characterised by an improperly positioned urethral opening, where the meatus is below the tip of the glans, and can be observed anywhere along the ventral side of the penis [7]. The severity of the hypospadias depends on the location of the urethral opening along the penis, with increasing severity as the meatus is closer to or at the scrotum. Severe forms often require surgery [23,24,25]. Hypospadias is a significant health problem and can mobilise considerable health care resources [26]. Indeed, more severe cases may require subsequent surgeries, due to complications, such as complete dehiscence, stenosis or fistulas [27].

Male urethral stricture is most commonly the result of an injury, including iatrogenic trauma, infection or non-infectious inflammatory conditions of the urethra. However, it can also be found after hypospadias surgery [28]. Less common causes can also be found, such as congenital urethral strictures or resulting from a malignant tumour. In the United States, office visits for urethral stricture reached nearly 1.8 million between 2005 and 2013, with a total annual cost of nearly $ 300 million in 2010 [29,30]. Urethral strictures can also lead to a high rate of urinary tract infection and incontinence [31].

### 2.3. Current Treatments

Repair or replacement of the urethra can be done using a wide variety of tissues, such as skin grafts, including genital and extragenital skin flaps, tunica vagina, lingual mucosa, bladder and mouth [32]. Oral mucosa has become the current gold standard; however, many complications are encountered. The first use of the oral mucosa as a urethral substitute was by Kirill Sapezhko, a Ukrainian surgeon from the Russian empire, in 1890 on a 40-year-old patient with an idiopathic urethral stricture [33]. This pioneering technique was also reported by Graham Humby in 1941 and independently rediscovered in 1992 by Bürger [34] and Dessanti [35]. However, all these substitutes have limitations compared to autologous urethral tissue, leading to complications, such as stenosis formation [36,37]. Despite the improvement of harvesting techniques, donor sites remain limited in the amount of tissue collected and can also be affected by complications. Indeed, about 80% of patients present side effects after harvesting buccal mucosa, such as pain, numbness, submucosal scars, dry mouth, lesions, neurosensory defects, discomfort and limitation of the opening of the mouth and risk of infections [38,39,40,41,42,43]. This lack of a sufficient amount of tissue to be grafted can be problematic in the case of extended defects. In addition, in postoperative complications, the surgeon cannot collect oral tissues twice from the same site, limiting surgical options.

## 3. Female Genitourinary Tissue: Focus on the Vagina

The vagina is an elastic muscular tubular structure that extends from 7 to 15 cm from the cervix to the vulva. Its primary roles are to allow sexual intercourse and to be the natural way for childbirth. It also has an essential role in the constitution of the bacterial flora of newborns, which poses the problem of the increase in births by caesarean section [44].

### 3.1. Anatomy

The vaginal wall consists of three layers: The vaginal mucosa, an intermediate muscle layer, and the adventitia, which serves as structural support for the organ. The vaginal mucosa is composed of a non-keratinised, multi-layered squamous epithelium [45]. The vaginal epithelium has an average thickness of 150–200 μm when the woman is premenopausal [46]. It has an original organisation with a basal layer, suprabasal layers, several middle layers of glycogen-rich cells and an apical layer [47]. Because the epithelial cells differentiate and migrate apically, their glycogen content increases. Thus, only the suprabasal and apical layers contain glycogen. The epithelial cells of the middle layers adjust their glycogen content depending on oestrogen levels. The vaginal epithelium varies throughout a woman’s life as it is subject to hormonal and environmental fluctuations. Before puberty, the vaginal epithelium is thin because it comprises only basal and suprabasal layers. When women are of reproductive age, the vaginal epithelium thickens and contains an average of 28 layers of epithelial cells [48]. After menopause, the decline in oestrogen concentration induces vaginal epithelial atrophy, depletion of glycogen reserves and a phenomenon of keratinisation on the surface of the vaginal epithelium, which becomes more similar to the epidermis [49]. The vaginal epithelium serves as a barrier to the entry of pathogens, mainly based on intercellular junctions between epithelial cells. The colonisation of the vagina by lactobacilli allows the production of lactic acid, which creates an acidic microenvironment, protecting against other potentially pathogenic bacteria and viruses, such as Human Immunodeficiency Virus (HIV) [50,51]. 

### 3.2. Pathologies

Numerous pathologies, whether congenital or acquired, can affect the vagina. Bladder and cloacal Exstrophy, in which the anatomical structures of the pelvis (including the bladder, genitals and colon) fail to fuse in the midline, are examples of such malformations [52]. Children with intersex disorders, such as congenital adrenal hyperplasia [53] and cloacal abnormalities, can have significant anatomical defects and often need outside tissue sources for reconstructive surgery. 

Müller’s agenesis, also known as Mayer–Rokitansky–Küster–Hauser syndrome (MRKH), is the most common abnormality in the development of the Müllerian duct, which results in vaginal agenesis [54,55,56]. MRKHS covers a range of abnormalities associated with vaginal and uterine abnormalities, but may also have other associated outcomes. 

Women with cancer of the cervix, uterus, ovary, rectum, vagina or bladder may require partial or total vaginal resection. Furthermore, they may need a partial or total vaginal replacement to recover sexual function and restore native anatomy. Vaginal stenosis can also occur after radiation therapy treatment to treat colorectal and cervical cancers (80% of women treated). It is characterised by a decrease in the length and diameter of the vagina, followed by the formation of scar tissue [57]. 

In addition, vaginal strictures can also occur with vaginal atrophy, hypoestrogenic states, inflammatory and autoimmune diseases, and chemical vaginitis [57]. Transient and long-term damage to the vagina and its supporting tissues have also been documented after vaginal birth. Most parous women have anatomical evidence of disturbed supporting tissues [58,59,60].

### 3.3. Current Treatments

Depending on the severity of the problem, non-surgical or surgical treatments can be proposed to the patients. Non-surgical treatments rely mainly on the Frank/Ingam procedure or the Vecchietti technique, while surgical treatments rely mainly on the Abbè/McIndoe procedure.

#### 3.3.1. Non-Surgical Treatments

Frank’s technique [61] involves using dilators, gradually increasing in diameter and length, which are placed three times a day in the future vaginal opening and held in place for 20 to 30 min. In Ingram’s technique [62], the patient uses a bicycle seat to hold the vaginal dilators in place, taking advantage of the pressure created by the patient’s weight. Both techniques require the presence of a vaginal dimple of 3–4 cm, which is not present in cases of severe agenesis. The advantages are the preservation of the existing vaginal tissue and the often-high satisfactory results. The limitations are the discomfort, the time required for this long-term treatment and the limitation to sexually mature patients [63]. The Vecchietti technique [64] is based on the placement of an acrylic olive connecting the vagina to the abdominal wall. The dilatation happens by internal traction. Within a few days, a neovagina is formed. The procedure is minimally invasive and preserves the existing vaginal tissue, but is limited by the discomfort and the use of dilators. There are risks of long-term contraction, prolapse and urological lesions.

#### 3.3.2. Surgical Treatments

Patients with congenital or acquired reproductive tract malformations often require extensive surgical reconstruction. In 1898, Abbè’s vaginoplasty was introduced, consisting of an autologous skin graft taken from the anterior thigh [65,66]. However, from 1938, autografts of skin for vaginal reconstruction became popular with the McIndoe method of wrapping a skin graft around a plastic stent. From the 1970s, the choice of grafts was oriented according to the areas of vascularisation, such as the musculocutaneous flaps of the gracilis muscle or the rectus muscle of the abdomen. This surgical procedure does not require dilators, making it a preferred procedure in a paediatric setting. Despite the difference of opinion on sigmoid vaginoplasty in the literature, this approach seems to give satisfactory results, but carries a risk of digestive complications. Other anatomical substitutes have been described for vaginal reconstruction: Oral mucosa, amniotic membrane, peritoneum, accelerated dermis matrices and oxidised cellulose matrices.

## 4. Tissue Engineering

In 1993, Langer and Vacanti published a paper in Science entitled “Tissue Engineering” (TE) [67]. This was the first assay to conceptualise the notion of TE and its goals. From this date, TE has been considerably developed to offer new and innovative solutions to old problems. In the context of ageing and the sedentary behaviour of human populations, the appearance of chronic diseases that require organ repair or replacement is increasing even though the number of organs available for transplants is reducing precisely because of this ageing and sedentary behaviour, but also the strengthening of selection criteria by regulatory agencies. Tissue engineering consists of the in vitro reconstruction of tissues or organs for the replacement/repair and 3D models for fundamental research. Synthetic or natural biomaterials, including decellularised organs as scaffolds, seeded or not with cells, have been studied. Various techniques have been described to create the scaffold, such as casting [68], electrospinning [69,70,71] or bioprinting [72]. Other methods like the self-assembly technique, which does not require any exogenous biomaterial, have also been developed.

### 4.1. Synthetic Materials

Synthetic biomaterials present many advantages, such as their low cost, ability to be highly tunable and form the desired geometry [73,74,75]. They also present some drawbacks, such as their difference in composition and organisation compared to a natural extracellular matrix (ECM), which can impact adhesion, proliferation and differentiation of cells [76]. Indeed, the low diversity of the components used to produce these biomaterials appears simplistic compare to the numerous components of a native ECM. Nevertheless, significant progress has been made with the functionalisation of synthetic biomaterials in the last decades. For many applications, the use of hydrogels is interesting, and recent studies have shown a potential for them functionalised with different peptides to recreate, at least in part, the microenvironment of the target tissues [77]. This kind of functionalised hydrogels could support cell growth/differentiation, as well as vehicles for the delivery of stem cells, drugs or bioactive proteins.

### 4.2. Natural Materials

Contrarily to synthetic materials, natural biomaterials used as scaffolds, especially those using elements coming from ECM [78], provide better adapted environments for migration, proliferation and differentiation of cells when used under appropriate conditions. Nevertheless, these biomaterials generally have weaker mechanical properties compared to native ECM. Indeed, the mechanical properties of ECM are not only due to the molecules it is composed of, but also, more importantly, because the organisation of these molecules in the matrix. 

Among other promising biomaterials, decellularised tissues are becoming more and more attractive with new protocols of decellularisation [79]. Indeed, it seems convenient to use the complexity of native materials to repair themselves. Decellularising tissues to remove antigenic molecules responsible for immunological response and potential pathogens before being recellularised with patient’s cells allows tissue production, which is very close to the autograft or even better if the disease’s causal effect is also corrected. Decellularisation of tissue can be obtained by successive hypo/hypertonic shocks, combined or not with the addition of detergents, to destroy the cell membrane [80]. Enzymes can be used to remove proteins and nucleic acids. The tissue can originate from a human cadaver or be obtained from animals.

Nevertheless, several problems slow down the use of such prostheses: Ethical problems and various regulatory issues, such as the risk of immune reaction in ineffective/incomplete decellularisation. However, the main problem remains the ability to maintain the ECM architecture that provides good mechanical and biological characteristics to the graft. Indeed, heavy decellularisation can induce a loss of the native matrix architecture and its non-immunogen components, whereas weak decellularisation could be insufficient to obtain a safe biomaterial [81,82,83]. In the past, ECM was largely disorganised or modified by the decellularisation process, losing the mechanical properties, resistance, elasticity and cell differentiation potential. An increasing number of protocols are emerging to reduce these problems, but more studies are needed to obtain a clear view for this option [84]. Recently a novel approach to functionalise decellularised tissues has been proposed to improve endothelial cell adhesion and accelerates endothelialisation, which is an important point, due to the need to rapidly provide oxygen and nutrients to grafts to ensure positive outcomes. This goal is achieved through selective immobilisation of REDV tetrapeptides [85].

### 4.3. Tissue Engineering for Urethral Reconstruction

Several groups have attempted urethral substitution using TE cell-free matrices, such as bladder acellular matrix graft (BAMG) and small intestinal submucosa (SIS) or cellularised matrices [86,87,88,89,90,91,92,93]. These matrices are prepared from native tissues by decellularising and sterilising them. As shown in rabbits by Dorin et al., a significant problem of acellular matrices is that urothelial regeneration is limited to 0.5 cm, which compromises success in more complex cases, such as long strictures [94]. Synthetic polymers have also shown advantages (poly-l-lactic acid, (PLLA) and poly(lactic-co-glycolic) acid, (PLGA)) for forming low-cost, biocompatible, three-dimensional (3D) organs with controlled mechanical properties. However, synthetic scaffolds without functionalisation by peptides do not allow the proper differentiation of epithelial cells into well-organised tissue. Indeed, contrarily to natural matrices, they cannot recreate the target organ microenvironment, especially adequate ECM-cell interaction (e.g., lack integrin-binding peptide sequence, failure in synchronisation between degradation rate and matrix neo-deposition) [73,95,96]. No long-term experiment has been performed with a significant number of patients. Currently, protocols developed are not used in clinics despite the media coverage of some, signalling the immaturity of the works, which must continue to be improved [97]. TE substitutes that contain autologous cells in addition to an extracellular matrix, close to the native one, are more promising. The main advantage of this method is that a large graft of autologous cells can be produced with a limited sample, such as a piece of oral or bladder mucosa. Indeed, the extracted cells can be grown in vivo, seeded on the biomaterial and implanted with a very low risk of rejection. Studies have also reported that stem cells can be obtained from urine, making this approach potentially useful [98,99]. A downfall of this method is that after long periods of culture to obtain well-differentiated tissues, the exogenous matrices become challenging to manipulate and lose their mechanical and physical properties. Despite significant progress in urethral TE, very few teams have performed clinical trials and published their results to date [100] (Table 1). However, the four clinical trials conducted to date show promising results in a limited number of patients with long-segment and/or complex stenosis [97,101,102,103,104]. While these models are certainly far from a “plug and play” alternative with consistently reproducible results, they could offer an alternative for complex cases requiring long segment urethral replacement [105]. 

### 4.4. Tissue Engineering for Vaginal Reconstruction

Vaginal abnormalities represent a significant health problem for women because nearly 1% of women will suffer from the pathologies mentioned previously, resulting in significant psychological impacts. Interestingly, tissue engineering is an area that aims to replace or regenerate dysfunctional tissues and organs with autologous cells, biomaterials or a combination of both. The success of vaginal reconstruction in these patients largely depends on the use of a sufficiently large tissue substrate that adequately fulfils the physiological functions of the vagina. Prior techniques have often relied on autologous tissues, such as the intestine or skin, often associated with complications, due to the physiological differences inherent to these substrates. To improve results, a variety of biodegradable substitutes, including collagen matrices and a decellularised bladder submucosa, have been used for vaginal replacement [102,134]. Reconstructions using these substitutes have generally failed, due to functional, structural, mechanical or biocompatibility issues. Using the patient’s vaginal tissue for reconstruction would be the most elegant and effective solution, but this has often not been possible, due to the relative scarcity of healthy vaginal tissue for autologous transplantation. There is a substantial clinical need to develop technologies to facilitate the regeneration of injured or diseased tissues and organs. The relentless prevalence of trauma, congenital abnormalities and diseases, such as cancer, is driving demand, becoming increasingly urgent as the world’s population grows and ages. A wide variety of tissues and organs would benefit from engineering-based repairing or regeneration. Several graft materials have been used to line the surgically created neovaginal cavity, including myocutaneous flaps or intestinal segments, full-thickness or split-thickness skin grafts, amniotic membrane, peritoneum, buccal mucosa and vaginal epithelial tissue [135,136,137,138,139,140,141,142,143]. These techniques are associated with contraction and/or stenosis of the graft, which may require long-term dilation. Oral mucosa vaginoplasties are associated with donor site morbidities, due to the large volume of tissue taken to create the neovagina. In addition, the amount of tissue that can be harvested from a donor site is limited, which can be problematic, especially for significant defects. To overcome these difficulties, alternative methods of vaginal reconstruction have been explored. Few groups have attempted TE vaginal reconstruction using acellular and cellular matrices of natural or synthetic origin [144,145,146,147,148,149] (Table 2). The tissues were transplanted into mice, rabbits and humans. However, further preclinical and clinical studies are needed, due to the limited number of subjects included in these studies. It remains challenging to determine whether the optimal technique was used.

## 5. The Self-Assembly Approach

New approaches are required to combine adequate cell signalling and differentiation, cell maintenance, especially for stem cells, and sufficient mechanical resistance for implantation. All this while minimising side effects. A new type of strategy has been explored by the group of Dr. François A. Auger at LOEX: The “self-assembly” method [6]. Great discoveries and therapeutic achievements have been made possible thanks to this unique technique which allows the production of reconstructed tissues free of exogenous materials. Indeed, the use of exogenous biomaterials can lead to immunological, foreign body reactions and the transmission of infections. This technique relies on cells cultured in ascorbic acid to secrete and deposit their own ECM to form cohesive sheets of cells and collagen [159,160]. While most biomaterials lose their mechanical and physical strength properties in culture, these self-assembled tissue properties are roughly similar or even exceeding those of native tissues in some models, due to the stabilisation of metalloproteinases [161]. The self-assembly technique has made it possible to construct models from various stromal cells of the skin [162], fat [163], cornea [164], Warton’s jelly [165], bladder [166] and vagina [157] exhibit not only excellent mechanical strength, but also an allowance for adequate epithelial differentiation. 

The first step of a tissue’s reconstruction using the self-assembly technique is to seed mesenchymal cells and cultivate them in the presence of ascorbate, also called vitamin C. It is preferable to use organ-specific mesenchymal cells (e.g., dermal fibroblasts to reconstruct skin substitutes, keratocytes to reconstruct cornea substitutes [167], bladder mesenchymal cells to reconstruct bladder mucosa substitutes [133] and vagina fibroblasts to reconstruct vaginal mucosa substitutes [168]). The use of unpaired mesenchymal cells can result in inadequate differentiation. Indeed, cutaneous differentiation of corneal or urothelial epithelial cells occurs when these cells are cultivated on dermal fibroblasts-derived stromas. The ascorbate concentration was set at 50 µg/mL even if it is not the optimal concentration for collagen deposition in dermal fibroblast cell culture. Indeed, higher concentrations of this oxidative agent induce cell death, and thus, reduce the total amount of deposited collagen and the mechanical strength of the final product. After 3 to 6 weeks, depending on the ability of the cultivated cells to deposit ECM, the mesenchymal cells have produced a thick material similar to a native stroma of the target organ (Figure 1). It is possible to refine the protocol by introducing, for example, at the initial seeding step, endothelial cells from an umbilical vein or, even better, from the organ-specific microvascular network to form a capillary-like network [169] or immune cells, such as monocyte-derived macrophages to produce immunocompetent models [168]. A reseeding step can also be done after two weeks to increase the tissue thickness and improve the cell distribution inside the stroma, which can especially be helpful for the microvascular network and increase elastic properties of the models [169] (Figure 2A). Following the step of stroma production, epithelial cells can be seeded on the top of the construct and cultivated for one week in submerged conditions, allowing the complete coverage of its upper surface before being placed at the air/liquid interface for three weeks to obtain differentiation of the epithelium. This technique has shown a high level of differentiation in various models, showing epitheliums very similar to native tissues. To place bladder tissues in physiological conditions, a bioreactor has been designed (Figure 2B). Detailed protocols can be found in several research articles and reviews [170].

For tubular substitutes, such as blood vessels or urethras, the epithelial/endothelial seeding does not happen immediately after the step of stroma production. The tissue-like structure is detached from the petri dish and tightly rolled around a mandrel of the appropriate diameter [161]. After the rolling step, the fusion of the rolls was helped by maintaining a mechanical load. The cylindrical mandrel can then be removed from the tubular structure, creating a lumen. The lumen is filled with liquid to avoid collapse, and epithelial/endothelial cells can be seeded inside the tube for urethral or blood vessel reconstruction, respectively. During the epithelial/endothelial cell seeding step, rotation of the tube ensures a uniform distribution of these cells (Figure 3A). Once again, urologic tissues can be matured under physiological conditions using bioreactors (Figure 3B). Various refinement has been introduced for these constructs. Notably, mesenchymal cells can be seeded directly on the mandrel to form the tubular structure, avoiding delamination of the rolls in the case of their incomplete fusion [173]. 

### 5.1. Self-Assembly to Reconstruct Human Tissues

Initially, the self-assembly technique had been developed to produce bilayered skin [162] for severely burned patients and its tubular derivation, the tissue-engineered blood vessel [161,177]. With time, the same technique has been used in a wide range of products useful in regenerative medicine, such as corneas [164,178], heart valves [179], adipose tissue [163,180,181], bone [182], bladder mucosa substitutes [166,171], ureter/urethra [174,176] and vaginal mucosa substitutes [157,158]. The most advanced substitute is the bilayered skin used to treat patients affected by burns and ulcers [183].

### 5.2. Self-Assembly Approaches to Produce Research Models

From these engineered tissues, has been derived, research models. In the case of the skin, bilayered skin substitutes were used to study skin [184,185] and uveal melanoma [186]. It allowed to study the effect of epidermal ultraviolet irradiation [187], the basal cell carcinoma [186] and neurofibroma [188], wound healing [189], hypertrophic scars [190,191], scleroderma [192], psoriasis [193] and epidermolysis bullosa [194]. They were also used to detect amyotrophic lateral sclerosis through the cutaneous manifestation of the disease [195]. Fat tissue substitutes are great models to study several aspects of metabolic disorders. Corneal substitutes are also exciting models to study Fuchs’ dystrophy [196] or eye wound healing [197]. The self-assembly technique is fascinating because ECM deposited by stromal cells recreates a microenvironment very similar to that of native tissue and reconstructs the modular models. For example, the contribution of the dermal fibroblasts has been demonstrated in psoriasis by combining different sets of fibroblasts and keratinocytes, healthy or diseased [193]. It was also demonstrated that fibroblasts extracted from skin biopsies of patients affected by scleroderma at an early stage of the disease remain sensitive to TGF-beta addition, increasing the dermis thickness of reconstructed tissues.

In contrast, the fibroblasts extracted from skin biopsies of patients at the late stage of the disease are insensitive to TGF-beta [192]. This finding can orient therapy for specific patients. It is also possible to add elements, such as capillary [169] or lymphatic [198] networks, immune cells [168] or microbiota, developing more refined models.

### 5.3. Self-Assembly Protocol for Urethral Substitute Model 

The self-assembly protocol was derived from producing urologic models. The skin model was adapted to produce a bladder mucosa substitute, whereas the blood vessel model was adapted to produce urethral substitutes. 

Initially, stromal and urothelial cells were extracted from a single porcine bladder biopsy [166]. It was shortly apparent that stromal cells of porcine origin—bladder, skin or oral mucosa—gave substitutes with inadequate mechanical properties [199]. It is simpler to obtain biopsies from human skin than the human bladder, thus, human dermal fibroblasts and porcine urothelial cells produce mechanically resistant and watertight substitutes [171]. Following this step, bioreactors’ maturation was tested to simulate the physiological pressure environment during bladder filling/emptying cycles [200]. After that, the importance of respecting the organ-specific pairing between mesenchymal cells and epithelial cells was demonstrated [133]. Human epithelial cells were used instead of porcine ones [172]. To simplify the process, the stacking of stroma sheets was replaced by a reseeding step which improves cell distribution throughout the tissue while still maintaining adequate mechanical properties [169]. This modification also forms a highly developed capillary-like network. Another source of easily obtained cells should be considered. Biomaterials derived from adipose stem/stromal cells (ASC) for bladder regeneration have been successfully designed [201]. The ASCs are easily harvested from a small sample of subcutaneous fat and yield a high proportion of multipotent cells, representing approximately 2%. They have immunomodulatory and proangiogenic properties that could potentially improve the quality of construction.

Nevertheless, reconstruction using these cells demonstrated that it should be avoided to put an ASC-derived stromal sheet directly in contact with urothelium. In direct contact, urothelial cells cannot differentiate adequately to form a robust basal lamina. However, interspersing one stromal sheet between the ASC-derived one and epithelial cells reestablished the ability of adequate urothelium maturation. Bladder mucosa substitutes could be used to replace/repair bladder after transurethral resection, i.e., cancer treatment. 

Similar steps were followed for urethral substitute reconstruction. Human dermal fibroblasts were used combined with porcine urothelial cells as a proof of concept to produce the first tubular structures [174]. Once the proof of concept was obtained, all cells were extracted from the human bladder and skin biopsies. A critical role of the liquid flow was demonstrated for an adequate differentiation of the urothelium compared to a static culture [176]. Tissue-engineered genitourinary tubular grafts were characterised for their histological and mechanical properties. This construction has an exceptional histological organisation and excellent mechanical resistance. Therefore, urethral substitutes were subcutaneously grafted in a mouse model to investigate the effects of prevacularisation [175]. An advanced 3D capillary-like network was observed. After transplantation, avoiding ischemic events is the main obstacle observed in developing a thick biomaterial or reconstituted tissue in vitro. Therefore, seeding endothelial cells can be used and adapted for human clinical applications. Indeed, the neovascularisation of a tissue is a slow process that can take more than 15 days for tissue that is 1 mm in thickness. The vascular network already presents in the conventional autologous transplant can reconnect by inosculation to the host’s bloodstream within four days. The solution to the revascularisation of reconstructed tissues in vitro could lie in the reconstruction of a capillary-like network in the graft before implantation. Encouraging results were obtained with inosculation on day 4 in the case of construct transplantation containing endothelial cells. In addition to improving the transplant, the early capillary network could also help remove blood cells, fibrin and growth factors, such as TGF-β1, leading to fibrosis and disease recurrence. It has, therefore, become obvious to use this tubular urological tissue as a model for urethral substitution. The advantage of this protocol using the self-assembly technique is that it contains mesenchymal cells that communicate with epithelial cells either by releasing cytokines and growth factors or by cell-to-cell contact. Although the best cellular source for urethral bioengineering by self-assembly is the mesenchymal cells of the patient’s target organ, this is impractical as there are risks associated with urethral biopsies, such as the creation of a fistula.

Nevertheless, it was demonstrated that cells derived from the bladder gave excellent results. The absence of exogenous materials and the autologous property of the models produced using the self-assembly protocols represent significant advantages over other available grafts. Therefore, an autologous TE urethra—without urothelium—was implanted in a rabbit model, which is the gold standard for the study of penile surgery [202]. Technical adjustments were necessary to produce substitutes using rabbit cells as culture conditions are different.

Several refinements were introduced in protocols to improve stem cell maintenance by expanding urothelial cells in hypoxia [203] or to shorten production time by adding lysophosphatidic acid in a cell culture medium [172]. 

Urethral abnormalities represent a significant public health issue as nearly 1% of men suffer from these pathologies, which can have a significant psychological impact. Treatment involves surgical correction, and current treatment options are associated with morbidities and a lack of long-term sustainable results. The solution may lie in reconstructing an autologous urethra from a small biopsy of the patient, the in vitro reconstruction, and its subsequent implantation. A purely autologous tissue would have a better outcome. The presence of stem cells during implantation would offer better growth potential, especially for paediatric patients. The urethral substitute produced by the self-assembly protocol is entirely autologous and free of exogenous material. It can be pre-endothelialised, and therefore, almost native histological and mechanical characteristics present before implantation. Thus, a reduction in adverse events following the transplantation of this living urethral tissue can be expected. Moreover, it is expected to grow and develop as the child ages. It would reduce the associated morbidity for patients at the implantation and harvesting sites, and therefore, decrease the financial burden of urethral abnormalities on the healthcare system.

Even if this model is associated with many advantageous characteristics, the graft preparation time is not negligible. From cell culture to full maturation, the reconstruction process takes three months, one month more if cell extraction/expansion is required. However, penile abnormalities are chronic pathologies, and surgical correction is performed on an elective basis. Indeed, most patients must wait months for surgery. Therefore, the extra time for surgical correction is not a significant inconvenience. The need for skin, bladder or fat biopsy is also a point to keep in mind, although this procedure is simple and minimally invasive for the skin and fat.

Nevertheless, in the future, induced pluripotent stem cells (iPSC) [204], which can be produced from the patient’s blood, could be differentiated into whole-cell types needed to reconstruct urethral substitutes. This new technology could avoid invasive procedures, such as harvesting by biopsy. Nevertheless, it is necessary to control the differentiation step for all types of cells used to avoid differentiation errors and the potential development of tumours. The significant expense associated with the manufacture of this biological graft material must be considered, keeping in mind that it would be completely autologous, a characteristic advantageous for patients. The current reference method is already associated with significant expenses, such as the frequent need for surgical intervention to be carried out in two stages, i.e., two distinct types of anaesthesia, several months apart, the additional risk of morbidity for the patient and the absence from work for a second major penile surgery (six weeks each time). The self-assembly approach to reconstruct urological tissue without biomaterial would open the door to reconstruct the ureters, spongiosum and cavernosal bodies of the penis, which would significantly improve the quality of life of patients severely disabled by congenital abnormalities. 

### 5.4. Disease Models Derived from the Urological Substitute Model

Modelling urological disease could be an asset in avoiding animal experimentation, and therefore, the interspecies differences that apply [205]. First, a bladder cancer model could help better understand its mechanisms, and therefore, finds more efficient treatments. The most common type of bladder cancer arises from the urothelium and results in urothelial cell carcinoma (UCC). Of all UCC, 70% of tumours are non-muscle-invasive confined to the urothelium and submucosa and associated with a favourable prognosis [206]. After transurethral resection followed by administration of immunotherapy or chemotherapy drugs, the UCC recurrence rate is the highest among solid tumours, reaching more than 70% [207,208].

Moreover, around 10–15% of such recurrent cases progress to invasive disease [209]. Understanding and preventing bladder cancer initiation and progression is of utmost importance in clinics. It is suspected that UCC secretes factors activating stromal resident fibroblasts into cancer-associated fibroblasts (CAF [210]). These cells could then modify the stroma to facilitate cancer progression [211]. Unfortunately, current in vitro models do not accurately recapitulate many critical aspects of cancer biology, especially 3D tumour-associated stroma and CAF activation [186]. The cancer stroma consists of an ECM populated by various cells, including endothelial cells, immune cells and fibroblasts. Under normal conditions, stroma acts by preventing epithelial-mesenchymal transition. However, when cancer starts to grow, the stroma switches towards a tumour-supportive function, primarily by CAF activation, and plays a crucial role in cancer progression [212]. To study these relations, a unique 3D UCC model based on tissue engineering has been recently developed [213] (Figure 4A). After normal urothelial cell seeding on the stroma, a mature urothelium developed on the basal lamina (appearing at day ten after air/liquid step), including an uroplakin positive umbrella cell layer. Spheroids from T24 (invasive) or RT4 (non-invasive) UCC cell lines were produced using the hanging drop method; they were implanted on the surface of the construct at day 10 (after basal lamina formation). Invasive and non-invasive behaviour of cancer cells from spheroids have been reported and open the way to use primary cancer cell populations. This 3D model can also be used to identify the molecular factors involved in the activation of fibroblasts into CAF.

Ketamine-induced cystitis has also been designed using urologic tissue substitutes. Ketamine is an anaesthetic widely used in human and veterinary medicine [214]. However, in recent years, it has also become a popular street drug under different names. Besides its surgical applications, especially in paediatric and veterinary medicine, ketamine is used to treat pain, mainly because it has little effect on respiratory and cardiac functions when used under intended conditions. This medicine has analgesic properties at doses lower than those used for anaesthesia [215]. Ketamine can also be used in the treatment of depression and certain addictive pathologies [216,217]. Of more significant concern, this product is also used beyond medical control for “recreational” purposes. Ketamine, inhaled or ingested in soluble form, is consumed in large quantities, which may cause serious health problems. In particular, since ketamine and its metabolites are mainly excreted in the urine, this drug causes severe inflammation of the urinary tract and bladder. These effects can lead to papillary necrosis, kidney failure and decreased bladder capacity secondary to chronic interstitial cystitis [218,219]. The main symptoms are urinary pain and burning associated with severe pollakiuria. Stopping ketamine improves symptoms, but there have been reports of permanent bladder size-reduction requiring bladder enlargement. Cultures of primary urothelial cells were treated with different increasing doses of ketamine in culture medium, and several parameters were evaluated: Growth curve, size and morphology of cells [220]. The doses used in these tests correspond to the doses found in the urine following therapeutic uses (0.5 and 1.5 mM) or (5 and 10 mM). The results clearly show a decrease in urothelial cell growth at 0.5 and 1.5 mM compared to the control (which may compromise the medium/long-term renewal of the bladder epithelium) and a decrease in the cell number at 5 and 10 mM (which suggests acute damage to the epithelium).

Likewise, increasing doses of ketamine induce a reduction in cell size at 5 and 10 mM. This very significant reduction is explained by the apoptotic morphology of the urothelial cells. Because apoptosis, or programmed cell death, is a mechanism caused by the activation of caspase enzymes, death by apoptosis of urothelial cells has been confirmed by assaying the activity of caspases. After that, 3D bladder mucosa substitutes have been used to study the toxic effect of ketamine (Figure 4B). Ketamine was applied on the mature urothelium using paper or agarose vectors for 48 h. Whatever the vector used, the macroscopic structure and the cell-cell cohesion of the urothelium, especially in the middle layers, were severely affected. In contrast, no effect was observed in adjacent, non-treated-areas.

The use of urethral stents to relieve urinary tract obstruction is constantly questioned because of the potential infection, encrustation and compression side-effects, which leads to the need for early removal procedures. Biodegradable ureteral stents, generally made from polymers, have been proposed to overcome these problems [221,222]. Recently, absorbable metals have been viewed as potential materials offering both biodegradation and resistance [223]. A model of urological tissue has been used to assess the use of such zinc-based alloys [224]. Histology of the reconstructed 3D ureter exposed to the metal showed a urothelium with characteristics close to the native tissue: Tight junctions were present at the surface layer and laminin at the basal layer, indicating a state of healthy tissue even with the presence of metal samples up to seven days of exposure.

### 5.5. Self-Assembly Protocol for Vaginal Substitute Model

Using a similar technique, a vaginal mucosa (VM) substitute has recently been produced [157,158] (Figure 5A). Because the best source of cells for VM bioengineered using the self-assembly technique is the patient’s target organ cells, epithelial and stromal cells were extracted from the human vagina. The reconstructed tissues exhibit excellent mechanical resistance and great elasticity. They had well-differentiated epithelium with oestrogen receptor beta expression, glycogen storage [157] and could be pre-endothelialised [158]. To assess the biocompatibility of the substitutes, subcutaneous grafts were realised in a mouse model. The tissue survived with no evidence of necrosis during the two months of in vitro reconstruction and three weeks after implantation. The same limitations and perspectives described for the urethral model can be applied to the vaginal model. The development of an autologous vaginal mucosa reconstructed by TE would be a significant advance in urogynaecology and would also have a considerable clinical impact. Providing a non-immunogenic, autologous and exogenous biomaterial-free substitute for the replacement of diseased tissues could bypass many disadvantages found when native tissues are taken from the patient—among them, avoiding the formation of scars after extraction is the primary benefit that can be found with the reconstruction of vaginal tissues using the self-assembly approach while using cells from patients. Surgical reconstruction using this method could significantly improve patients’ quality of life and could potentially reduce the financial burden of vaginoplasty on the healthcare system. 

### 5.6. Disease Models Derived from the Vaginal Substitute Model

Investigating specific molecular events happening during infection of the human vagina by HIV or fungi, such as *Candida albicans*, is limited by the lack of appropriate experimental models [205,225]. Animal models especially show species-specific differences, such as lactobacillus colonisation, hormonal cycle or vaginal epithelium organisation. New biologically relevant models of the human female reproductive tract are needed to develop new efficient drugs to treat patients. Therefore, models useful to investigate infectious pathologies were derived from the human 3D organ-specific vaginal bilayered mucosa reconstructed for vagina replacement/repair. A published example is HIV infection [168]. The model proved to be histologically close to native tissue, hormone-responsive, offer mechanical resistance to be used in vitro, and sustained a macrophage population infected or not by HIV-1.

Using human primary untransformed organ-specific cells, i.e., vaginal fibroblasts for the stroma reconstruction and vaginal epithelial cells for the epithelium, allows adequate crosstalk between the stroma and the epithelium, ensuring the production of a histologically similar native tissue architecture. The model presents the proliferation and maturation of the vaginal epithelium showing glycogen amounts that decline with the patient’s age from whom the cells were extracted. To mimic the hormonal cycle, which induces in vivo variation in the thickness of the vaginal epithelium rendering it more susceptible to infection when thinner, a 10 nM supply of oestrogen was used during the reconstruction of the vaginal mucosa of a 32-year patient. Oestrogen addition increases epithelial thickness as expected, but has no impact on the tissue’s mechanical properties.

Once the primary model was characterised, it was challenged for HIV infection. No active viral replication was observed when free HIV-1 particles were delivered in an immuno-incompetent model, i.e., without the presence of monocyte-derived macrophages (MDM) in the model. No more active viral replication was observed if MDMs HIV-infected was seeded on the apical side of the 3D reconstructed vaginal mucosa tissue. However, viral replication and transcytosis were observed when immunocompetent 3D reconstructed vaginal mucosa tissues incorporated MDMs in the stroma and were infected with free HIV-1 GFP viral particles.

This vaginal mucosa model offers a physiologically relevant tool to explore viral load and HIV-1 transmission in an environment that may contribute to propagating the virus and may help to develop new antiviral treatments in vitro. This model has also been used to mimic an infection of the vaginal mucosa by *Candida albicans*, but the results are not yet published. 

## 6. Perspectives

The field of tissue engineering can bring potential alternatives to the constantly increasing need for tissue repair/replacement of urological organs. To do so, a real challenge is to combine good mechanical properties while maintaining stem cell potential. It must also allow adequate epithelium differentiation, and its degradation product should not imply adverse effects. It must be vascularised easily and not represent an immune risk for patients, while being functional as soon as the graft is in place. We can, therefore, better understand the significant obstacle that must be overcome. 

In this article, we detailed the self-assembly technique, without using synthetic or decellularised biomaterials/scaffolds. Indeed, genitourinary tissues have been developed using animal cells and have been successfully implanted in animals. To facilitate clinical translation and avoid interspecies differences, the next step is to graft human organ-specific tissues in immunosuppressed animals before trying the prototype on human subjects. These tissues constitute a promising avenue for the surgical correction of various defects, whether from congenital or acquired origin.

Several new challenges emerge, such as urothelial, mesenchymal and endothelial cells differentiated from iPSC derived from blood cells. This would avoid the need for a biopsy with its potential comorbidities, and avoid the problem of the non-organ-specific communications between cells leading to aberration or the presence of inadequate cells to harvest (e.g., cancer). However, there remains a lot to be done to obtain an adequate differentiation from iPSC without aberrant expression of oncogenes. Another challenge in the coming years should be to exclude the use of serum in cell culture. Serum could induce a loss of reproducibility in some experiments and be a source of contamination and raise ethical concerns. Furthermore, the world’s increase in demand will keep dragging the price up while the global production of serum is stagnant.

## Figures and Tables

**Figure 1 bioengineering-08-00099-f001:**
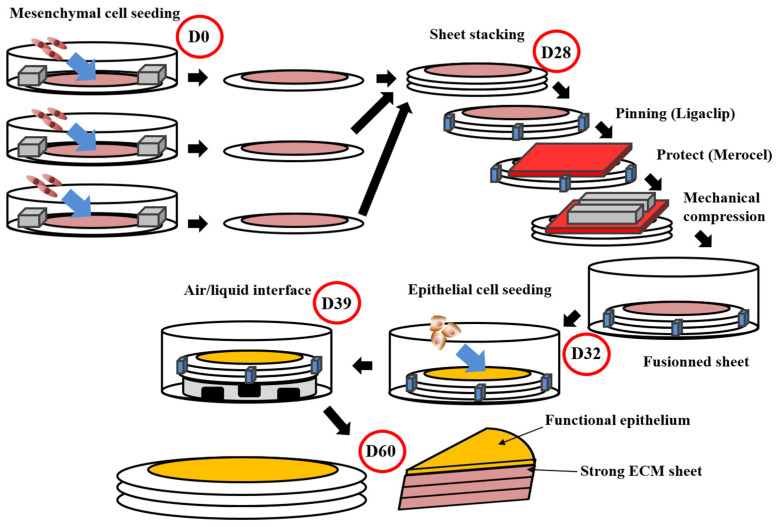
Schema of the self-assembly method using the standard protocol. This protocol obtains flat tissues with functional epithelium and a strong ECM sustaining manipulations by surgeons or researchers [166,171,172]. The total time of production around 60 days. For the whole process, media are supplemented with ascorbate (50 µg/mL). On day 0 (D0), mesenchymal cells are seeded into petri dishes containing a paper ring as an anchorage device weighted by small metal lingots. Cells are then cultured for 28 days. On day 28 (D28), stroma sheet (cells + ECM) are stacked (air bubble must be avoided between sheets), then they are pinned together using a surgical clip (Ligaclip), covered by a surgical sponge (Merocel) to protect from the direct contact with metal lingots use to favour fusion through mechanical compression. Culture is pursued for 4 days until day 32 (D32) to ensure adequate fusion. Then epithelial cells are seeded on the top of the construct and cultured for a week to allow horizontal coverage of the scaffold. On day 39 (D39), the constructs are raised at the air/liquid interface using a specific device allowing media circulation under the reconstructed tissue. This phase allows the maturation of epithelium until day 60 (D60).

**Figure 2 bioengineering-08-00099-f002:**
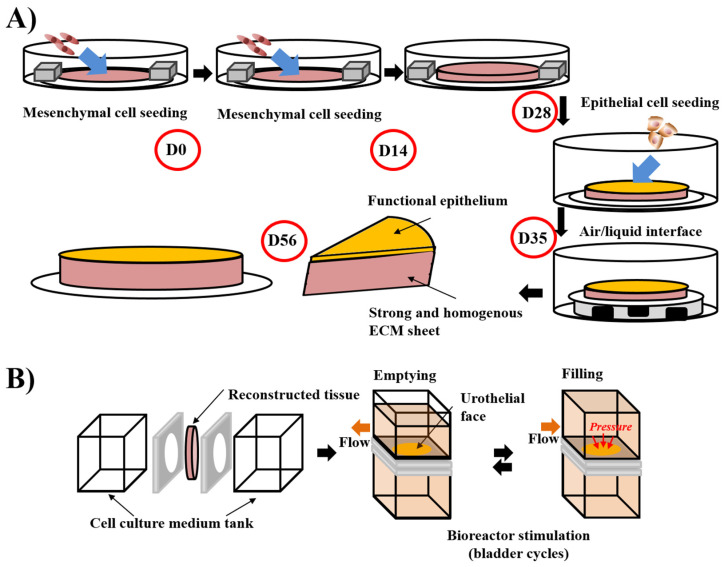
Schema of the “reseeding” variation of the self-assembly protocol and the stimulation of bladder mucosa substitute in a bioreactor. (**A**) The “reseeding” protocol reduces the cost and the complexity of the self-assembly technique compared to the standard protocol, avoiding the need for multiple sheets and their stacking. A better distribution of the cells throughout the tissue has also been demonstrated. (**B**) Stimulation of bladder mucosa substitute using bioreactor. The substitute is inserted between two chambers where the cell culture medium can circulate. To mimic the bladder cycles of emptying and filling, the flow is modified and increased the pressure on the urothelial face of the substitute in the filled status, whereas the substitute is more relaxed in the empty condition.

**Figure 3 bioengineering-08-00099-f003:**
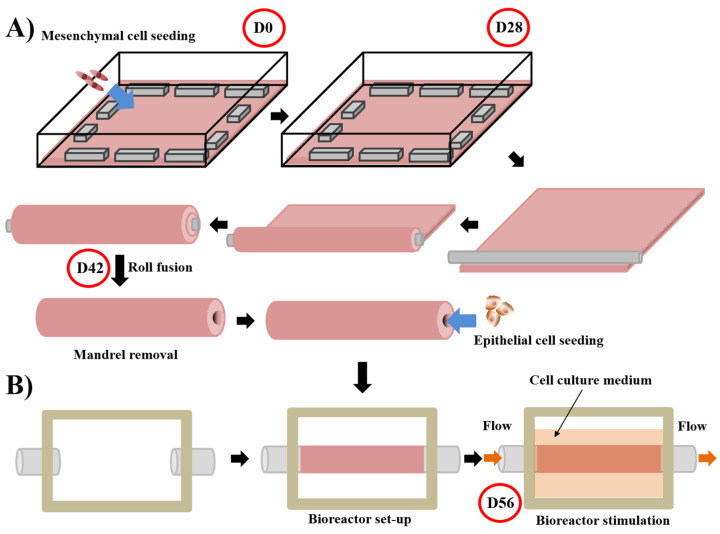
Schema of the technique used to produce tubular substitute using the self-assembly approach and the subsequent stimulation in a bioreactor of the substitutes. For the whole process, media are supplemented with ascorbate (50 µg/mL). (**A**) Mesenchymal cells are seeded at day 0 (D0) on a gelatin-coated square plate and cultured for 28 days. Contraction of the tissue is avoided by putting flat metal lingots on its border. After the mesenchymal cells formed an ECM sheet (day 28 (D28)), the latter was rolled tightly around a cylindrical mandrel. Fusion of the rolls is allowed by maintaining a little load on the tissue for 14 days. On day 42 (D42), the mandrel can be removed and epithelial cells seeded inside the tubular structure [174,175]. (**B**) The day after, a bioreactor separates the external and internal parts of the tissue, a flow circulating in the internal part mimicking the physiological flow. The cell culture medium is present in the external part to provide nutrients. Maturation is done for 14 days until day 56 (D56) [176].

**Figure 4 bioengineering-08-00099-f004:**
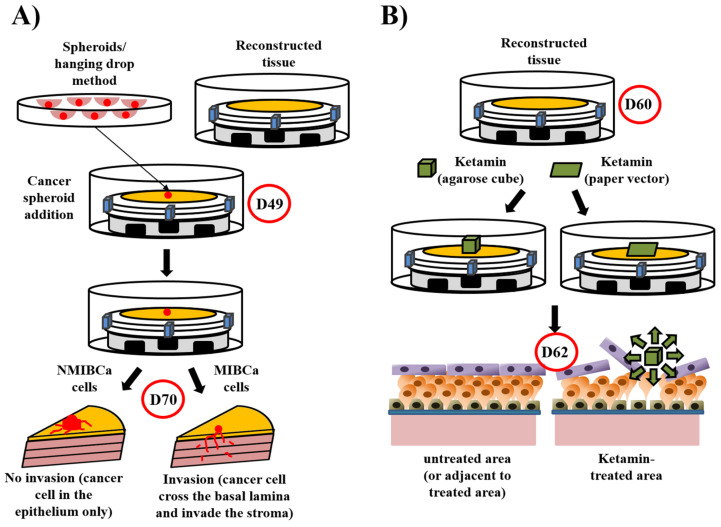
Models derived from the bladder mucosa substitute: Bladder cancer model and ketamine-induced cystitis. (**A**) The bladder cancer model. The tissue were reconstructed until the 10th day of air/liquid interface step (day 49). Bladder cancer cell spheroids were produced in parallel using the nagging drop technique. On day 49 of tissue production, just after the basal lamina is formed, spheroids were added and the culture is pursued 21 days more. As expected, spheroids made from non-muscle-invasive bladder cancer cells (NMIBC, RT4 in detailed experiments) remained in the epithelium and form papillary structures, whereas muscle-invasive bladder cancer cells (MIBC, T24 in detailed experiments) crossed the basal lamina and invaded the stroma. (**B**) The ketamine-induced cystitis model. Reconstructed tissue (day 60) was treated with agarose cube containing ketamine of paper imbibed of ketamine. In both cases, severe damages were observed in the area treated, but not in adjacent areas or in untreated conditions. The superficial layer of the affected area was disturbed and intermediate layers mainly destroyed.

**Figure 5 bioengineering-08-00099-f005:**
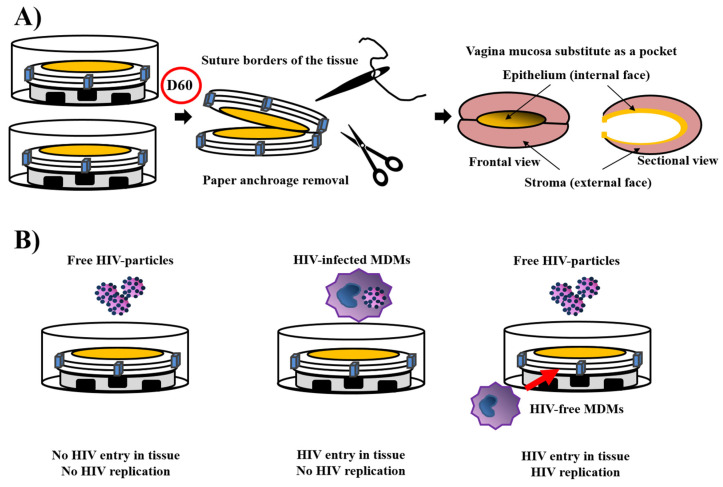
Reconstruction of vaginal mucosa pockets and the use of vagina mucosa as a HIV infection model. (**A**) The vagina mucosa pocket. Two reconstructed vagina mucosa were associated, their epithelium facing each other. The paper anchorage was removed, while the two substitutes were sutured. An opening is let to allow stent entry and obtaining of a physiological air/liquid condition. Such pockets were subcutaneously grafted into mice with success. (**B**) The HIV infection model. After reconstruction of vagina mucosa three situations were examined, free HIV particles on the epithelium, HIV-infected monocyte-derived macrophages (MDM) on the epithelium, or free HIV particles on the epithelium, but with MDMs into the stroma. Only this latter condition showed the entry of HIV in tissue and its replication.

**Table 1 bioengineering-08-00099-t001:** The scaffold used for urethral reconstruction.

Type of Scaffold	Biomaterials	Ref Example	Advantages	Drawbacks
**Synthetic**	PLCL	[106]	- biocompatible - mechanical properties	- Degradation products
	PLCL/Collagen	[107]	- low cost	- Poor differentiation of epithelial cells (except for cellularised collagen matrices; improved by functionalisation)
	PLA	[108]	- highly reproducible	-degradation rate (too low or too high)
	PU/mesh in PGA	[109]	- quickly available	-mechanical properties during or after degradation
	PLGA	[97]	- functionalisation	- poor angiogenesis
	PLLA	[110]		
**Natural**	Cellulose	[111]		
	Silk Fibroin	[86,112,113,114]		
	Collagen	[78,88,115,116,117,118,119]		
**Acellular matrix**	SIS	[81,120,121,122,123,124,125]	- Adequate microenvironment for cell proliferation and differentiation	- Immune risk (including DNA, prions)
	Placental membrane	[126]	- Significant angiogenesis	- Unfavourable clinical experience
	BAMG	[127,128,129]		- Quality of the matrix
	Urethra	[130]		
**Self-Assembly**	None	[73,131,132]	- Excellent microenvironment with organ-specific cells - Mechanical properties	- time and cost to produce tissues

Most of the data in Table 1 can also be found in reviews [73,131,132,133]. PLCL: poly(l-lactide-co-ɛ-caprolactone); PLA: polylactic acid; PU: polyurethane; PGA: polyglycolic acid; PLGA: poly(lactic-co-glycolic) acid; PLLA: poly-l-lactic acid; SIS: small intestinal submucosa; BAMG: bladder acellular matrix graft.

**Table 2 bioengineering-08-00099-t002:** The scaffold used for vaginal reconstruction.

Type of Scaffolds	Biomaterials	Patients #	References
Synthetic	PGA	4	[149]
	PLA (©PACIENA)	9	[150]
		7	[151]
Natural	Collagen IV and hyaluronic acid	1	[146]
		23	[146]
Acellular matrix	Amnion	50	[152]
	SIS	65 (vs Interceed)	[153]
		Monkey	[154]
	Acellular vaginal matrix	Rat	[155]
		Rat	[148]
Artificial dermis		35	[156]
Self-Assembly		Mouse	[157,158]

Patient # = number of patients included in the study. PGA: polyglycolic acid; PLA: polylactic acid; SIS: small intestinal submucosa.

## Data Availability

Not applicable.

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
