# Peer review of "Genitourinary Tissue Engineering: Reconstruction and Research Models"

_bioengineering, 2021, doi:10.3390/bioengineering8070099_

Round 1

Reviewer 1 Report

In this review, Caneparo et al. reviewed male and female genitourinary tissue engineering.

First, they gave an outline of the anatomy, pathologies, treatments of male and female ganitourinary diseases. Then, genitourinary tissue engineering, which is expected to be a new treatment for genital diseases were reviewed in detail.

This paper provides scientifically correct information based on a wide range of literature and is expected to provide useful information to the readers of this journal. Therefore, my decision is that his manuscript is acceptable after minor modifications.

1. All figures should be quoted in the text.

2. It would be useful information if the cell scaffolds, type of cells, target diseases, etc. are summarized in a table.

Reviewer 2 Report

I think this review is a well written, interesting, and useful contribution for genitourinary area, which I think is suitable for publication in "bioengineering".

In my opinion it requires minor revision before ready for publication. Now therefore, there is several details listed below that the authors should address.  

1) The authors describe that the synthetic scaffolds do not allow the proper differentiation of epithelial cells into well-organized tissue. but in the case of artificial blood vessels, there are many articles that have been carried out without any problem even in the synthetic polymer. Please cite the references to describe the poor organization of the epithelium tissue on synthetic polymer scaffolds such as PLLA and PGA.  

2) Figure 1 caption doesn't explain of the figure. I can't understand what they mean. Please explain the figure more detail by the caption. Moreover, please cite the references to this scheme. Also, it can be said in the title of the diagram, but please specify what "D" in the diagram describes.  

3) In Figure 3, the method of tube formation is shown, but it is described in the protocol which is common to urinary path and blood vessel. It is also related to comment 1, but can we think about the epithelial cells and the endothelial cells together in this methods? I think Is it better to narrow it down to a urology. Moreover, please cite the references in this protocol.

Reviewer 3 Report

The proposed review appears very well structured and organized. 

Anyway, some little suggestion should be useful in improving the quality of the work.

The authors should collect some of the listed issues (where it is possible) in form of tables, allowing the readers to instantly and graphically collect the information. This is because the authors reported progresses in fields other than those described in the title.

Furthermore, regarding decellularized tissues, the authors could have a look at a very interesting work proposed by Dal Sasso et al. (Bioactive materials 2021) regarding a novel approach for the decoration of decellularized tissues thus improving endothelial adhesion, promoting obtained scaffolds for blood-contacting applications.

Finally, in the context of self assemby approach, the work proposed by Zamuner et al (Materials 2016) could provide an interesting starting point in the design and development extracellular matrix (ECM) analogues, 3D support for cell growth as well as vehicles for the delivery of stem cells, drugs or bioactive proteins.

Figures should be cited in the main text.

References seems correctly reported in the main text.
